

# Self-esteem and stress: a structural equation modelling of biosocial determinants, psychological mediators and anxiety among Malaysian undergraduates

Muhammad Ikhwan Mud Shukri, Anisah Baharom and Ahmad Iqmer Nashriq Mohd Nazan

Department of Community Health, Universiti Putra Malaysia, Serdang, Selangor, Malaysia

Corresponding author
Anisah Baharom,
anisbaharom@yahoo.com

## ABSTRACT

Anxiety is a widespread public health concern affecting youth worldwide, leading to significant functional and social disabilities, various negative social and financial consequences, and even suicidality. However, there is a lack of research examining the roles of self-esteem and stress in mediating factors contributing to anxiety among undergraduate students, particularly in Malaysia. Therefore, this research aims to investigate these determinants and the mediating effects of self-esteem and stress on the relationship between biosocial factors and anxiety among Malaysian undergraduates. A cross-sectional study was conducted with 1,193 undergraduates using a validated questionnaire. The study examined independent variables including gender, sleep quality, problematic internet use, social support, and mental health literacy; mediators (self-esteem and stress); and anxiety as the dependent variable. After excluding 68 potential outliers, the final structural equation model demonstrated satisfactory goodness of fit (root mean square error of approximation (RMSEA) = 0.041, $\chi^2$/df = 2.918, and comparative fit index (CFI) = 0.925). Mediation analysis using the bootstrapping method revealed that self-esteem and stress fully mediate the relationship between gender, problematic internet use, social support, and anxiety. Mental health literacy was found to be fully mediated by self-esteem alone. Both self-esteem and stress partially mediated the relationship between sleep quality and anxiety. The full mediation structural model accounted for 63.8% of the variance in anxiety. Interventional efforts targeting anxiety among undergraduates could significantly benefit from strategies aimed at enhancing self-esteem and mitigating stress. Future research should explore the levels of emotional social support and its association with anxiety among undergraduate students.

## INTRODUCTION

Anxiety is widespread among university undergraduates, with a global prevalence exceeding 30% (*Batra et al., 2021*; *Wenzhen et al., 2022*). A recent study across 16 Malaysian

universities involving 1,821 students revealed a similarly high prevalence of anxiety, at 29% (*Mohamad Sidik, Akhtari-Zavare & Gani, 2021*). Notably, university students are more prone to severe and chronic illnesses if they experience anxiety or mood issues early on (*Mofatteh, 2021*). The recurrence of symptoms with each episode escalates the likelihood of future anxiety episodes, increasing the risk of long-term anxiety disorder development (*Scholten et al., 2023*). Consequently, young individuals grappling with anxiety often face a myriad of adverse health and social repercussions, including academic struggles, strained relationships with peers, behavioural challenges, conflicts with authority figures, substance misuse, poor physical health, compromised work performance in adulthood, and an elevated risk of suicide (*Hauenstein, 2003*; *Lew et al., 2020*; *Keenan-Miller, Constance & Brennan, 2007*).

Anxiety is defined by persistent intrusive thoughts and concerns that cause ongoing worry and tension (*Tan et al., 2023*). Similarly, it is also described as excessive worry and fear disproportionate to everyday situations, leading to negative thoughts and predictions about future events (*Santabarbara et al., 2021*). Several risk factors contribute to anxiety among undergraduates, encompassing various domains, including (1) biological factors such as gender and insufficient physical exercise; (2) social factors such as socioeconomic status, social change issues, poor social support, low mental health literacy, problematic internet use, smoking, alcohol consumption, illicit drug use, prior mental health problems, poor sleep quality, relationship issues; and (3) psychological factors such as low self-esteem and stress (*Attlee et al., 2022*; *D'Hondt et al., 2020*; *Hamaideh, Modallal & Tanash, 2022*; *Huang et al., 2021*; *Kok & Low, 2019*; *Ma et al., 2020*; *Ramón-Arbués et al., 2020*; *Yi et al., 2017*).

Gender differences in mental health are influenced by socialization and gender-specific experiences. For instance, men are more frequently diagnosed with substance abuse or antisocial disorders, while women are more prone to depression and anxiety. Women often experience internalizing symptoms like loneliness and withdrawal, whereas men tend to exhibit externalizing symptoms such as aggressiveness and impulsivity (*Eaton, 2011*). Several studies highlight that female students are at a higher risk of anxiety (AOR: 2.23, 95% CI [1.56–3.17]) compared to their male counterparts (*D'Hondt et al., 2020*; *Kessler et al., 2015*; *Ramón-Arbués et al., 2020*), which may be linked to factors like perceived body image and academic performance (*Gao, Ping & Liu, 2019*). Women are more likely to dwell on negative emotions through repetitive rumination rather than engaging in problem-solving activities, which contributes to their higher prevalence of anxiety. However, findings regarding the association of gender with anxiety are varied. Some studies suggest no statistically significant association (*Fauzi et al., 2021*; *Sun & Zoriah, 2015*), potentially due to limitations in sample size among male populations.

Sleep quality refers to an individual's satisfaction with their sleep experience, encompassing factors like sleep efficiency, latency, duration, and wake after sleep onset (*Kathy, Davis & Corbett, 2022*). Good sleep quality has beneficial effects such as feeling fully rested, restoring bodily functions, and promoting positive mental well-being. Conversely, poor sleep quality is linked to symptoms like fatigue, irritability, daytime dysfunction, and slow cognitive responses. However, current knowledge gaps exist regarding the mechanisms

through which sleep impacts mental health (*Columbia University, 2022*). Research indicates that university students with poor sleep quality are twice as likely to experience anxiety (AOR: 2.15, 95% CI [1.52–3.03]) compared to those with good sleep quality (*Ramón-Arbués et al., 2020*), a finding supported by studies conducted in the US and Ethiopia (*Dagnew, Andualem & Dagne, 2020*; *Wilson et al., 2014*). Recognizing the importance of maintaining good sleep quality could help students reduce psychological distress.

Problematic internet use (PIU) is characterized by a maladaptive fixation with internet usage, leading to excessive time spent online, persistent use despite negative consequences, and impaired functioning without any other underlying pathology to account for these behaviours (*Aboujaoude, 2010*). Essentially, it entails excessive internet use resulting in adverse outcomes, often requiring clinical judgment to diagnose addiction. Among university students, PIU can stem from various causal pathways. For instance, digital communication between students and parents is commonplace and has become increasingly prevalent over the years. Platforms like Facebook facilitate virtual interactions between students and their families. While parental oversight can serve as a protective factor against PIU at home, monitoring internet usage becomes more challenging when students are away at university. Coupled with the necessity for parent–child digital communication, this lack of supervision heightens the risk of PIU among university students (*Kerr et al., 2020*). PIU significantly impacts the mental well-being of university students. Studies have indicated that students with PIU are nearly three times more likely to experience anxiety compared to those without PIU (*Ramón-Arbués et al., 2020*).

Social support involves providing assistance or comfort to others to help them cope with biological, psychological, and social stressors. This support can come from various sources such as family, friends, colleagues, or structured support groups (*Harandi, Taghinasab & Nayeri, 2017*), and may include physical, financial, or emotional forms of aid. According to the social causation model, low social support can contribute to psychological distress by affecting self-esteem and fostering negative thoughts. For instance, a cross-sectional study in China found that university students who perceived low social support were nearly six times more likely to experience anxiety (AOR: 5.98, 95% CI [5.75–6.23]) compared to those with high perceived social support (*Ma et al., 2020*). Transitioning to university life often involves spending less time with established family and friends, which can be a source of psychological stress. Establishing social support among new peers at university takes time, particularly for incoming freshmen who may initially perceive lower support compared to their earlier social environments. Factors such as limited time spent on campus during the first year can further restrict access to social networks and diminish feelings of belonging (*McLean, Gaul & Penco, 2022*).

Mental health literacy (MHL) refers to an individual's knowledge about recognizing, managing, and preventing psychological distress and illnesses (*Kutcher, Wei & Coniglio, 2016*). It includes understanding various aspects such as recognizing psychological distress and illnesses, identifying risk factors and causes, knowing self-help interventions, accessing professional help, fostering positive attitudes towards seeking help, and understanding how to find mental health information (*Sampaio, Gonçalves & Sequeira, 2022*). University students with low MHL may struggle to identify psychological distress early, leading

to prolonged illness duration and delayed treatment. They may also face challenges in accessing mental health services and applying self-help strategies to manage their distress. Research has consistently shown that low MHL is significantly associated with an increased risk of anxiety among university students (AOR: 3.68, 95% CI [2.86–4.72]) (*Chi, Thai & Nguyen, 2018*; *Huang et al., 2021*). Conversely, studies have also highlighted the protective effect of good mental health literacy in reducing anxiety (AOR: 0.02, 95% CI [0–0.61]) (*Ying et al., 2022*).

Self-esteem plays a crucial role in safeguarding the mental well-being of undergraduates. It bolsters their confidence in learning and striving for academic excellence, fosters the setting of high goals, and provides resilience in dealing with challenging tasks or setbacks (*Du, King & Chi, 2017*). Enhancing self-esteem is paramount in maintaining the mental health of undergraduates. At its core, self-esteem is rooted in self-acceptance. It involves adopting a positive attitude towards oneself and one's characteristics, signifying an individual's embrace of their own identity (*Qian, Yu & Liu, 2022*). Low self-esteem is one of the risk factors for anxiety among university students with varying magnitudes (*Nguyen et al., 2019*; *Ramón-Arbués et al., 2020*). Many theories such as Beck's cognitive theory postulated that negative belief about one self, which is the essence of low self-esteem, contribute to the development of mental illness (*Beck, 1967*). Low self-esteem affects undergraduates both cognitively and motivationally. Those with low self-esteem tend to hold negative views about themselves and may have avoidance motivation to shield themselves from mental health stressors. Conversely, undergraduates with high self-esteem are motivated to pursue and maintain positive self-regard. These differing motivational tendencies influence goal-setting behaviours (*Hankin, 2006*). Cognitively, individuals may believe that approval from society, friends, and peers is necessary for happiness, leading them to worry excessively about potential negative events in the future. This can result in feelings of sadness, depression, and disappointment, further fuelling negative self-judgment and diminishing self-confidence. The perception of disapproval from others can exacerbate feelings of guilt, anxiety, and distress (*Masselink, 2018*).

Current literature lacks prevalence and trend analyses of low self-esteem among university students in Malaysia. A study conducted among undergraduates in African countries revealed that approximately one fifth of the respondents reported experiencing low self-esteem, with prevalence ranging from 19.0% to 21.4% (*Horesa et al., 2021*; *Paudel et al., 2020*). The high prevalence of low self-esteem underscores the importance of implementing intervention programs targeting the enhancement of self-esteem among university students. Integrating self-esteem strategies into mental health interventions can serve a protective role against the development of poor mental health, particularly through the promotion of higher levels of self-efficacy. This, in turn, fosters students' belief in their ability to successfully accomplish tasks and pursue goals (*Mohammadzadeh et al., 2018*). Despite the significance of these findings, there remains a dearth of studies exploring the prevalence and trends of low self-esteem specifically within the context of Malaysia. Thus, emphasizing the critical need for the present study to fill this gap in knowledge.

On the other hand, a study conducted in a Universiti Putra Malaysia (UPM) revealed that approximately one-fourth of medical undergraduates experienced stress (*Minhat &*

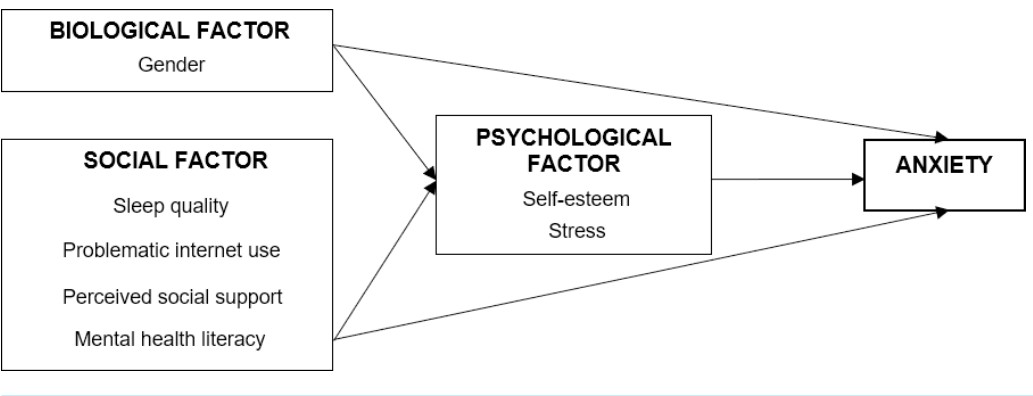

**Figure 1  The conceptual model.**

*Alawad, 2019*). This prevalence is consistent with the global pooled prevalence of stress among university students, which stands at 26.0%, as reported in a recent meta-analysis (*Batra et al., 2021*). Previous research has demonstrated a significant relationship between stress and anxiety various factors that predict anxiety among undergraduate students (*Hamaideh, Modallal & Tanash, 2022*; *Yu et al., 2022*). Identifying these predictors is crucial for guiding researchers in developing interventions to address anxiety. However, there is a limited body of research examining the roles of self-esteem and stress in mediating the relationship between predictors and anxiety among undergraduate students. Past studies have explored the roles of self-esteem and stress in mediating relationships between predictors and other mental health illnesses, such as depression and schizophrenia (*Kim & Jang, 2019*; *Lin, 2015*; *Al Nima et al., 2013*). Besides, researchers have extensively investigated the roles of self-esteem and stress in mediating psychological factors, including personality traits and depression, in relation to anxiety. However, there is a gap in the literature regarding the roles of self-esteem and stress in mediating biological and social factors with anxiety among undergraduate students. In recent years, a growing number of new social factors associated with anxiety have emerged, including problematic internet use, which aligns with the advancements in technology and increased internet usage (*Poorolajal et al., 2019*). Exploring the roles of self-esteem and stress in mediating these social factors is crucial as it elucidates the primary mechanisms through which these factors contribute to anxiety. Moreover, there is a scarcity of studies investigating the mediating effect of self-esteem and stress within theoretical frameworks. Therefore, this study offers valuable insights into the predictors and mediators of anxiety, enhancing our understanding of the causal pathway leading to anxiety beyond mere associations between variables.

Drawing from the Biopsychosocial Theory (*Engel, 1977*), our hypothesis posited that biological and social factors could lead to anxiety, mediated by psychological factors among undergraduates. The conceptual model developed in this study, as illustrated in Fig. 1, delineates the relationship between biosocial factors (such as gender, sleep quality, problematic internet use, social support, and mental health literacy) and anxiety, mediated by psychological factors (namely, self-esteem and stress).

## METHODS

### Study design and sample size

The study was conducted among undergraduate students in 14 faculties in Universiti Putra Malaysia (UPM), one of the public universities in Malaysia. Employing a cross-sectional study design, data collection occurred from June 19, 2023 to August 25, 2023. Proportionate random stratified sampling was utilized to select the required number of students from each faculty. To determine the sample size, the ratio of undergraduate students in each faculty was calculated based on a list of 15,340 undergraduate students enrolled in the first semester of 2022/2023, obtained from the university's admission department. The sample size calculation was conducted using Soper's online power analysis tool, considering a statistical power level of 0.8, a probability level of 0.05, an effect size of 0.15, seven latent variables, 69 observed variables, and an adjustment for a non-response rate of 40%. Consequently, the required sample size was determined to be 1,305 undergraduates. All students enrolled in undergraduate programmes were included in the study, except those who had participated in the study's pretest or had deferred their enrolment in undergraduate programmes.

### Study instrument

This study utilized a validated self-administered questionnaire, which was available in both Malay and English languages. The questionnaire comprised two sections: Section A included sociodemographic variables such as age, gender, and ethnicity, while Section B covered sub-sections on sleep quality, problematic internet use, social support, mental health literacy, self-esteem, stress, and anxiety.

Sleep quality was measured using the validated Sleep Quality Scale (SQS), which comprises seven items, known for their strong psychometric properties with internal consistency coefficient of 0.72 (*Önder et al., 2016*). It assessed various aspects of sleep, including sleep latency, nighttime awakenings, sleep quality perception, and feeling rested after sleeping. Each item was scored from 1 to 3, with a higher total score indicating better sleep quality (note: the 1st, 2nd, 3rd, 4th, and 7th items were reverse scored).

The Problematic Internet Use Questionnaire-6 (PIUQ-6) was employed to assess problematic internet use in this study. This scale consists of six items and has demonstrated robust psychometric properties, achieving a Cronbach's alpha of 0.82. It retains three domains from the original PIUQ: obsession, neglect, and control disorder, while reducing the number of items within each domain to two (*Göktaş et al., 2018*). Each domain comprises two questions, resulting in a total of six questions in the scale. Responses are rated on a 5-point scale ranging from one (indicating no problematic internet use) to five (indicating very frequent problematic internet use). The total score ranges from six to 30, with higher scores indicating a greater likelihood of problematic internet use.

Multidimensional scale of perceived social support (MSPSS) questionnaire was used to measure perceived social support (*Song et al., 2023*). This scale comprises of 12 items, each rated on a 7-point Likert scale. It measures support from family, friends, and significant others, with each domain consisting of four items. All three domains demonstrated good internal consistency with Cronbach's alpha ranging from 0.90 to 0.93. Scores on the MSPSS

range from 12 to 84, with higher scores indicating greater perceived social support among participants. Alternatively, the total score can be divided by 12, and the average score categorized into levels of support: low support (scores one to 2.9), moderate support (scores three to five), and high support (scores 5.1 to seven).

The Mental Health Literacy Scale (MHLS) questionnaire was utilized to assess mental health literacy (*Campos et al., 2022*). The Cronbach's alpha for the scale was 0.84. This scale comprises 16 items, each rated on a 5-point Likert scale. It encompasses four dimensions: (1) knowledge of mental health problems (six items), (2) erroneous belief/stereotypes (three items), (3) help-seeking and first aid skills (four items), and (4) self-help strategies (three items). Scores on the MHLS range from 16 to 80, with higher scores indicating greater mental health literacy. Notably, three items in the erroneous beliefs dimension (Items two, five, and six) were reverse-scored.

Self-esteem was assessed using the Rosenberg Self-Esteem Scale (RSES), consisting of ten items (*García et al., 2019*). Half of these items are expressed as positive statements, while the other half are negative statements. The internal consistency coefficient for the scale was 0.75. Prior to analysis, five items (item numbers 2, 5, 6, 8, and 9) were reverse-scored. Responses were recorded on a 4-point scale (0 = strongly disagree; 1 = disagree; 2 = agree; 3 = strongly agree), yielding total scores ranging from 0 to 30. Higher RSES scores indicate higher levels of self-esteem. Participants' self-esteem levels were categorized as follows: scores 0 to 14 (low self-esteem), scores 15 to 25 (normal self-esteem), and scores 26 to 30 (high self-esteem).

Stress levels were assessed using the Perceived Stress Scale-10 (PSS-10) questionnaire as it demonstrated good internal consistency coefficient of 0.78 (*Harris et al., 2023*). This tool consists of ten items and measures the global perception of stress by evaluating feelings and thoughts over the past month. Each item is scored from 0 to 4, resulting in total scores ranging from 0 to 40. Higher scores indicate higher levels of stress. Stress severity was categorized as follows: scores 0 to 13 (mild stress), scores 14 to 26 (moderate stress), and scores 27 to 40 (severe stress).

Anxiety was assessed using the Generalized Anxiety Disorder-7 (GAD-7) questionnaire, a validated tool utilized in both primary care settings and the general population with excellent Cronbach's alpha of 0.92 (*Sapra et al., 2020*). This questionnaire consists of seven items, each rated from 0 to three, indicating the frequency of experiencing different anxiety symptoms over the past two weeks. Response options include "not at all," "several days," "more than half the days," and "nearly every day," corresponding to scores of 0, 1, 2, and 3, respectively. Total scores on the GAD-7 range from 0 to 21, with higher scores indicating higher levels of anxiety. Additionally, a score of 10 or higher suggests the respondent is at risk of experiencing clinically significant anxiety.

The questionnaire was provided in both English and Malay languages. The initial questionnaire was in English, then it was translated into Malay using a back-to-back translation process to ensure accuracy and consistency. Two bilingual public health physicians performed the initial translation from English to Malay, followed by back-translation by two primary care physicians proficient in both languages. To validate the

translation, the questionnaire was then submitted to the Centre for the Advancement of Language Competence (CALC) at UPM for final confirmation.

## Pretest

Following the translation process, a pretest was conducted with 30 UPM students from the Faculty of Veterinary who were not part of the main study. Their role was to evaluate the face validity of the questionnaire by providing feedback on its structure and wording. These students were excluded from the actual data collection process. Their feedback indicated that no modifications to the questionnaire were necessary. Furthermore, two public health physicians validated the content validity of the questionnaire. Each item which relates to the four independent variables, two mediators, and the dependent variable recorded a content validity ratio (CVR) value of 1, indicating unanimous agreement on the essential nature of all items. As a result, all items were retained in the questionnaire. Additionally, the Cronbach's alpha coefficient for all variables ranged from 0.51 to 0.95, as observed from the pretest. This range signifies acceptable internal consistency reliability across the measures utilized in the questionnaire.

## Data collection and analysis

Data collection was carried out from June to August 2023. The management section of each faculty was approached, briefed on the research and permission was obtained to distribute the questionnaire. The questionnaires were distributed in stages to the 1,305 selected undergraduates with the assistance of the head student from each faculty. Respondents were encouraged to respond to all items in the questionnaire honestly. These questionnaires were collected within a one-week period. Prior to participation, all students provided informed consent using official written documents. Ethical approval was obtained from the Ethics Committee for Research Involving Human Subjects of Universiti Putra Malaysia (JKEUPM) (Reference number: JKEUPM-2023-164).

Data entry and analysis were conducted using IBM SPSS (version 25; IBM Corp, Armonk, NY, USA). The distribution of data was assessed using the Shapiro–Wilk test, Kolmogorov–Smirnov test, and evaluation of kurtosis and skewness. For multivariate analysis, structural equation modelling (SEM) was employed. In the present study, the SEM analysis comprised confirmatory factor analysis (CFA), measurement model, structural model and mediational analysis. In the CFA stage, the factor loadings of each item, the Cronbach's alpha, the average variance extracted (AVE) and composite reliability of each latent variable were examined. Subsequently, the measurement model analysed each latent variable in the model to be identified and measured by the observed variables (items). This model was crucial for assessing discriminant validity, multicollinearity, and multivariate normality.

The confirmed measurement model from the CFA was combined with a manifest variable, specifically gender ($=0$ for men, $=1$ for women), to construct the structural model, illustrating the relationships between variables. The structural model was assessed for overall model fit, size, direction, and significance of each path analysis, ensuring the fitness of the model. This study employed nine indices categorized into three fit

index criteria for model evaluation: (1) Parsimonious fit, includes chi-square ($\chi^2$/df); (2) Absolute fit, includes significance level (*p*-value), root mean square error of approximation (RMSEA), and goodness-of-fit index (GFI); and (3) incremental fit, includes adjusted GFI (AGFI), comparative fit index (CFI), Tuker Lewis index (TLI), incremental fit index (IFI), and normal fit index (NFI). Finally, a bootstrapping method was employed for mediational analysis. In this approach, the data underwent resampling 5,000 times with a 95% confidence interval for bias-corrected option. A syntax-based, user-defined estimand function was used to assess the mediation effect of self-esteem and stress individually for each variable simultaneously.

# RESULTS

Out of the 1,305 respondents initially selected, a total of 1,193 undergraduates participated in the present study, yielding a response rate of 91.4%. The prevalence of anxiety among participants was found to be 18.7%. Table 1 presents the descriptive results of the biosocial determinants, self-esteem, stress and anxiety. The mean age of respondents is 22.2 (1.4); with majority of respondents were female (73.0%) and of Malay ethnicity (73.2%). The mean score of sleep quality was 15.3 (2.5) and median score of mental health literacy was 69.0 (10.0). Additionally, a significant proportion reported high problematic internet use (85.3%) and high social support (51.0%). Furthermore, the majority exhibited high self-esteem (72.6%) and moderate levels of stress (76.2%), while 81.3% did not report experiencing anxiety.

Subgroup analyses were conducted to assess PIU, social support, and mental health literacy across respective domains. Regarding PIU, findings indicated that among the domains studied, the mean score for control disorder was the highest ($3.0 \pm 0.8$), followed by neglect ($2.9 \pm 0.8$) and obsession ($2.6 \pm 0.9$) (see Table S1).

For undergraduates with high social support, significant others provided the highest mean support score ($6.0 \pm 0.8$), followed by family ($5.8 \pm 0.9$) and friends ($5.6 \pm 0.9$). Conversely, undergraduates with low social support scored lowest in support from significant others ($1.8 \pm 0.9$), compared to family ($2.5 \pm 1.0$) and friends ($2.8 \pm 0.9$).

Additionally, further analysis revealed that the mean score for social support increased with each subsequent year of study (Year $1 = 59.2 \pm 1.4$, Year $2 = 60.4 \pm 1.5$, Year $3 = 61.4 \pm 1.5$, Year $4 = 62.3 \pm 1.5$, Year $5 = 63.2 \pm 1.6$). An ANOVA analysis indicated a significant difference among the years of study ($F = 2.524$, $p = 0.039$); specifically, social support differed significantly between Year 1 and Year 4 (see Table S2).

In the first stage, CFA was performed with item-level analysis was conducted for each latent variable. Meanwhile, parcel-level analysis was conducted for problematic internet use, social support and mental health literacy according to their respective domains. For social support, one item (item eight) exhibited a very low factor loading of 0.007 ($p = 0.811$) and was consequently omitted from further analysis. Table 2 presents the results of each test conducted in this stage. All three parcel-level model were chosen to be included into analysis as they possessed better AVE value compared to item-level model while still maintaining acceptable values for Cronbach's alpha and composite reliability.

**Table 1 Distribution of biosocial determinants, psychological mediators and anxiety among UPM undergraduate students ($N = 1,193$).**

| Variables | Mean (SD) | Frequency | Percentage (%) |
|---|---|---|---|
| Biological factor | | | |
| Age (years) | 22.2 (1.4) | | |
| Gender | | | |
|    Male | | 322 | 27.0 |
|    Female | | 871 | 73.0 |
| Ethnicity | | | |
|    Malay | | 873 | 73.2 |
|    Chinese | | 168 | 14.1 |
|    Indian | | 93 | 7.8 |
|    Others | | 59 | 4.9 |
| Social factor | | | |
| Sleep quality | 15.3 (2.5) | | |
| Problematic internet use | 17.2 (4.3) | | |
|    High ($\geq$13) | | 1018 | 85.3 |
|    Low (<13) | | 175 | 14.7 |
| Social support | 60.5 (12.8) | | |
|    High (>60) | | 608 | 51.0 |
|    Moderate (36–60) | | 540 | 45.2 |
|    Low (<36) | | 45 | 3.8 |
| Mental health literacy | 69.0 (10.0)[*] | | |
| Mediator | | | |
| Self-esteem | 33.4 (7.0) | | |
|    High ($\geq$30) | | 865 | 72.6 |
|    Moderate (26–29) | | 164 | 13.7 |
|    Low ($\leq$25) | | 164 | 13.7 |
| Stress | 19.9 (5.6) | | |
|    Mild (<14) | | 146 | 12.2 |
|    Moderate (14–26) | | 909 | 76.2 |
|    Severe (>26) | | 138 | 11.6 |
| Anxiety | 6.0 (5.0)[*] | | |
|    No (<10) | | 970 | 81.3 |
|    Yes ($\geq$10) | | 223 | 18.7 |

**Notes.**

SD, Standard deviation.

*Median (IQR).

In the second stage, all latent constructs were simultaneously entered to the model without assignment to exogenous or endogenous variables. The findings of the measurement model showed that the model did fit the indices. The relative chi-square was 3.698, the GFI = 0.882 , AGFI = 0.864 , CFI = 0.902, IFI = 0.902, NFI = 0.871, TLI = 0.893, RMSEA = 0.048 and AIC = 3296.565. Eight items (three items in sleep quality, one item in MHL and four items in stress) had factor loadings less than 0.5. However, they were retained in the model because factor loadings above 0.3 were considered acceptable,

**Table 2  Confirmatory factor analysis.**

| Variables | Cronbach's alpha (CA) | Average variance extracted (AVE) | Composite reliability (CR) |
|---|---|---|---|
| SQ | 0.700 | 0.253 | 0.687 |
| PIU | | | |
| IL | 0.781 | 0.356 | 0.765 |
| PL | 0.754 | 0.515 | 0.760 |
| SS | | | |
| IL | 0.914 | 0.390 | 0.874 |
| PL | 0.724 | 0.471 | 0.727 |
| MHL | | | |
| IL | 0.853 | 0.271 | 0.850 |
| PL | 0.676 | 0.409 | 0.715 |
| SE | 0.881 | 0.475 | 0.887 |
| Stress | 0.826 | 0.320 | 0.802 |
| Anxiety | 0.915 | 0.605 | 0.914 |

Notes.

SQ, Sleep quality; PIU, Problematic internet use; SS, Social support; MHL, Mental health literacy; SE, Self-esteem; IL, Item-level analysis; PL, Parcel-level analysis.

particularly given the significantly large sample size (more than 300) (*Hair et al., 1998*), and because the overall model fitness was achieved despite these lower factor loadings (*Awang, 2012*).

Table 3 displays the results of the discriminant validity assessment based on the values of average variance extracted (AVE) and squared correlation coefficient ($r^2$). Among the latent constructs, 16 pairs demonstrated discriminant validity, as indicated by AVE values exceeding the squared correlation coefficient ($r^2$). However, five pairs of latent constructs did not meet this requirement, namely: (1) Sleep Quality (SQ)—Self-Esteem (SE), (2) SQ—Stress, (3) SQ—Anxiety, (4) SE—Stress, and (5) Stress—Anxiety. Despite not meeting the typical AVE > $r^2$ criterion, the correlation coefficient ($r$) values for these five pairs of latent constructs were all below 0.8, ranging from 0.541 to 0.757. This aligns with criteria established by other researchers who suggest thresholds above 0.8 (*Rönkkö & Cho, 2022*) or 0.9 (*Voorhees et al., 2016*) as indicating discriminant validity issues. Thus, the discriminant validity of the measurement model, based on our study results, was acceptable.

The assumption of multivariate normality was assessed using skewness and kurtosis measures. The skewness values for all items was within the normal value of $-3$ to $+3$ (ranging from $-1.480$ to $1.368$), while kurtosis values for all items were also within the normal value of $-7$ to $+7$ (ranging from $-1.007$ to $2.889$), as recommended (*Schumacher & Lomax, 2012*). Hence, the measurement model was considered to be normally distributed. To check for outliers in the measurement model, high Mahalanobis d-squared ($d^2$) values were examined, with both p1 and p2 values being 0.000. A total of 68 cases were identified as potential outliers, scoring high Mahalanobis $d^2$ values. Consequently, these cases were excluded from the dataset, resulting in 1,125 responses available for further analysis

**Table 3** Composite reliability and discriminant validity of latent variables.

| Variables | CR | SQ | PIU | SS | MHL | SE | Stress | Anxiety |
|---|---|---|---|---|---|---|---|---|
| SQ | 0.687 | 0.252 | | | | | | |
| PIU | 0.760 | 0.144 | 0.514 | | | | | |
| SS | 0.724 | 0.205 | 0.017 | 0.467 | | | | |
| MHL | 0.729 | 0.017 | 0.006 | 0.162 | 0.417 | | | |
| SE | 0.903 | 0.293[*] | 0.099 | 0.429 | 0.118 | 0.509 | | |
| Stress | 0.829 | 0.301[*] | 0.195 | 0.138 | 0.004 | 0.543[*] | 0.349 | |
| Anxiety | 0.914 | 0.328[*] | 0.118 | 0.125 | 0.002 | 0.329 | 0.573[*] | 0.604 |

Notes.

The AVE of constructs (on the Diagonal) and squared correlation coefficient ($r^2$) (on the off-Diagonal).

SQ, Sleep quality; PIU, Problematic internet use; SS, Social support; MHL, Mental health literacy; SE, Self-esteem.

[*]Indicates $r^2$>AVE.

(1193−68). The assumption of multicollinearity of measurement model was evaluated using correlation coefficient ($r$) between latent constructs. All pairs of latent constructs scored $r$ value less than 0.9, indicating absence of multicollinearity in the measurement model (*Hair et al., 1998*).

In the third stage, the structural model was developed according to the proposed relationship outlined in the conceptual framework. Following the removal of 68 potential outliers, the analysis was conducted using a dataset comprising 1,125 respondents. Figure 2 depicts the final structural model, illustrating the relationship between biosocial determinants, psychological mediators and anxiety with standardized regression estimates.

Sensitivity analysis was conducted to robustly examine various structural models using individual items instead of their aggregated constructs. In Model 2, the analysis employed individual items from the PIU questionnaire for both the measurement and structural models, rather than using the PIU construct as a whole. Similarly, in Model 3, social support items were used directly in the analysis, and in Model 4, MHL items were similarly utilized during these analyses. Table 4 illustrated the findings of final model against other models. Based on the sensitivity analysis result, Model 1 portrayed the best model fitness with the lowest AIC as compared to the other models.

In the final stage, the bootstrapping method was employed to investigate possible mediation effect of self-esteem and stress on the relationship between biosocial determinants and anxiety. A total of 5,000 bootstrap samples were selected, with a 95% confidence interval for the bias-corrected option. The results of the mediation analysis, presented in Table 5, revealed that self-esteem fully mediated the relationship between gender, problematic internet use, social support, and mental health literacy with anxiety. Conversely, stress fully mediated the relationship between gender, problematic internet use and social support with anxiety. Additionally, both self-esteem and stress partially mediated the relationship between sleep quality and anxiety. This study demonstrated that due to high self-esteem, female undergraduates experienced lower levels of anxiety, while due to high stress levels, female undergraduates experienced higher levels of anxiety. However, stress did not mediate the relationship between mental health literacy and anxiety. The full mediation structural model accounted for 63.8% of the variance in anxiety, while the

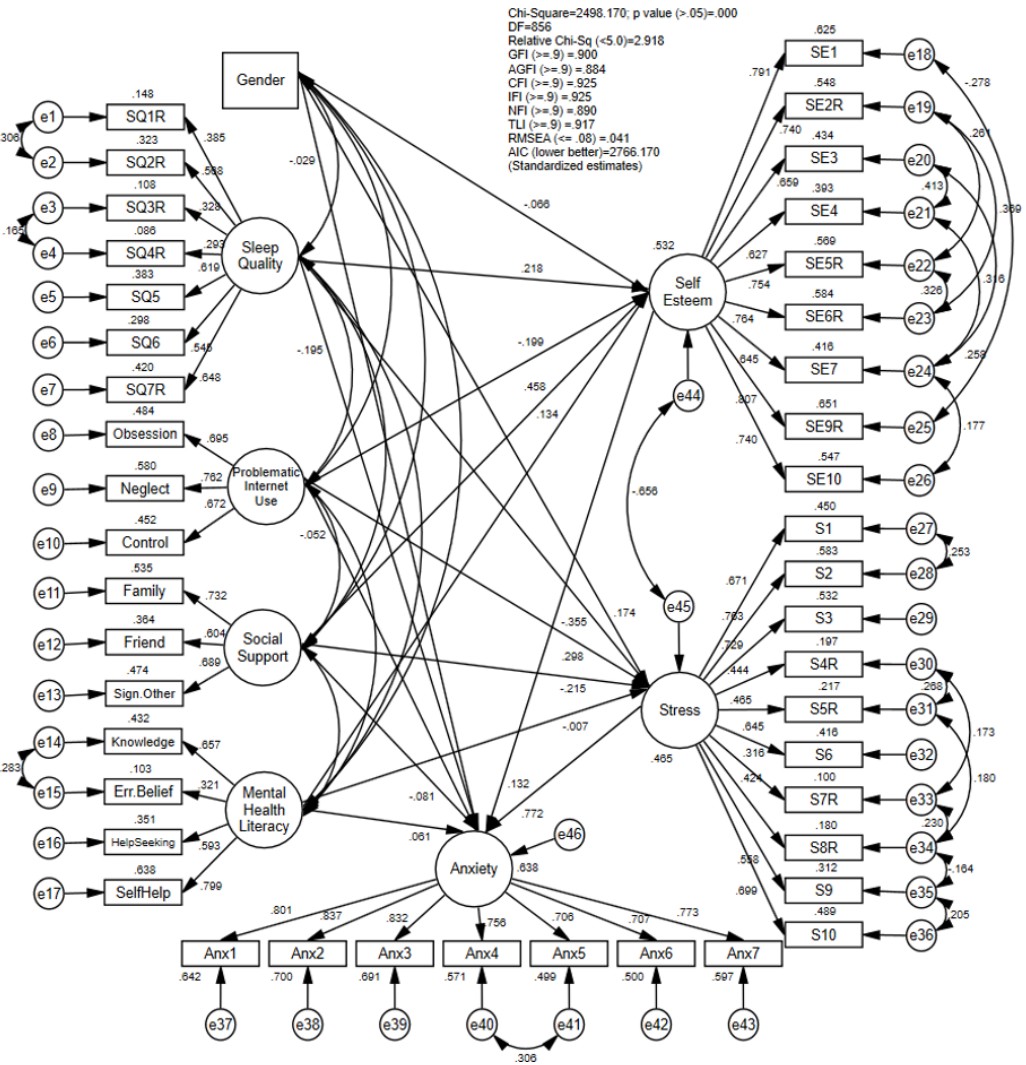

**Figure 2** The final structural model with standardized estimates.

direct structural model only explained 37.6% of the variance. This finding suggests that self-esteem and stress substantially contributed an additional 26.2% to the variance in anxiety.

## DISCUSSION

The present study revealed a relatively low prevalence of anxiety among undergraduate students, at 18.7%. This finding contrasts with previous literature, which reported higher prevalence rates of anxiety among university students in Malaysia (29%) (*Mohamad Sidik, Akhtari-Zavare & Gani, 2021*), as well as a global meta-analysis among university undergraduates (39.65%) (*Ahmed et al., 2023*). However, the result is comparatively higher than a local study which recorded a prevalence of anxiety among university students at 8.0% (*Sundarasen et al., 2020*). Both local studies involved large samples from public and

**Table 4  Sensitivity analysis of different structural models.**

| Goodness of fit measures | χ²/df | RMSEA | GFI | NFI | CFI | TLI | AIC |
|---|---|---|---|---|---|---|---|
| Recommended fit indices | <5.0 | ≤0.08 | ≥0.9 | ≥0.9 | ≥0.9 | ≥0.9 | The lower, the better |
| Model 1 (Final Model) | 2.918 | 0.041 | 0.900 | 0.890 | 0.925 | 0.917 | 2,766.170 |
| Model 2[*] | 2.930 | 0.041 | 0.891 | 0.879 | 0.916 | 0.906 | 3,175.293 |
| Model 3[*] | 3.518 | 0.047 | 0.867 | 0.866 | 0.900 | 0.892 | 4,788.013 |
| Model 4[*] | 2.932 | 0.041 | 0.869 | 0.850 | 0.895 | 0.888 | 4,528.719 |

Notes.

$\chi^2$/df, Relative Chi-Square; RMSEA, Root Mean Square of Error Approximation; GFI, Goodness of Fit Index; NFI, Normal Fit Index; CFI, Comparative Fit Index; TLI, Tucker-Lewis Index; AIC, Akaike Information Criterion.

[*]In Model 2, individual PIU questionnaire items were used instead of the full construct. Similarly, Model 3 used social support items, and Model 4 used MHL items directly in the analyses.

**Table 5  The indirect effect of biosocial determinants on anxiety through self-esteem and stress.**

| Hypothesized path | Direct effect (b) | Indirect effect (b) | 95% Bootstrap BC CI | | p | Mediation |
|---|---|---|---|---|---|---|
| | | | LB | UB | | |
| Gender⟶ Anxiety | −0.029 | | | | 0.226 | |
| Self esteem | | −0.012 | −0.037 | −0.001 | 0.023 | Full |
| Stress | | 0.189 | 0.129 | 0.263 | <0.001 | Full |
| Sleep quality⟶ Anxiety | −0.195 | | | | <0.001 | |
| Self esteem | | 0.065 | 0.006 | 0.164 | 0.032 | Partial |
| Stress | | −0.624 | −0.904 | −0.416 | <0.001 | Partial |
| Problematic internet use⟶ Anxiety | −0.052 | | | | 0.120 | |
| Self esteem | | −0.025 | −0.058 | −0.002 | 0.032 | Full |
| Stress | | 0.218 | 0.149 | 0.300 | <0.001 | Full |
| Social support⟶ Anxiety | −0.181 | | | | 0.074 | |
| Self esteem | | 0.040 | 0.002 | 0.087 | 0.038 | Full |
| Stress | | −0.109 | −0.169 | −0.058 | <0.001 | Full |
| Mental health literacy⟶ Anxiety | 0.061 | | | | 0.064 | |
| Self esteem | | 0.034 | 0.003 | 0.086 | 0.031 | Full |
| Stress | | −0.010 | −0.137 | 0.113 | 0.867 | No |

Notes.

BC, Bias-corrected; 5,000 bootstrap samples.

private universities nationwide. However, the significant difference in prevalence may be attributed to students' living arrangements. During the COVID-19 pandemic, when the latter study was conducted, the majority of students stayed with their parents, which likely provided feelings of safety and belonging, potentially reducing anxiety.

In the present study, one potential explanation for the variance in prevalence could be attributed to the timing of data collection, which coincided with examination weeks. During this period, students were residing in university accommodations, which may have contributed to a higher prevalence compared to previous studies (*Sundarasen et al., 2020*). However, during examination weeks, undergraduate students often experience heightened

stress levels. They tend to seek and receive strong social support from family and friends, as indicated by the study findings. Social support plays a vital role in fostering resilience and social skills among university students, enabling them to access support networks during stressful situations (*Broks et al., 2022*). Previous research has demonstrated that resilience aids in reducing anxiety levels by promoting positive life perspectives, accessing social resources, and employing specific coping strategies (*Harandi, Taghinasab & Nayeri, 2017*). Moreover, greater access to social resources, such as robust family and friend networks, instils feelings of love, care, and belonging among university students. Effective communication within these networks serves as a buffer against anxiety (*Ghasemipoor & Jahanbakhs, 2010*). By utilizing these coping mechanisms, students exhibit increased resilience, leading to more positive adaptations to anxiety during stressful periods (*Fullerton, Zhang & Kleitman, 2021*). Besides, variations in anxiety screening practices using different screening tools may contribute to disparities in prevalence between the present study and previous research. Sub-analyses in pooled anxiety meta-analyses have shown variations in anxiety prevalence among university students based on different assessment tools such as DASS-A (52.10%), GAD-7 (30.24%), and BAI (36.29%) (*Ahmed et al., 2023*), a trend supported by a recent systematic review (*Shukri et al., 2023*).

The study revealed that self-esteem and stress fully mediated the relationship between gender and anxiety among UPM undergraduates. There is a significant gap in research regarding the role of self-esteem and stress as mediators for gender. Previous literature has highlighted the role of hormonal imbalance in the development of anxiety among women. Serotonin levels, known to improve self-esteem, mood, sleep, and appetite, have been associated with lower anxiety levels in numerous studies (*Nishizawa et al., 1997*). Conversely, previous literature has indicated that stress can cause changes in estrogen and cortisol levels in humans (*Verma, Balhara & Gupta, 2011*). Higher cortisol levels have been detected during stressful situations (*Kajantie & Phillips, 2006*). We suggest that female undergraduates may be more sensitive to the effects of increasing cortisol levels during stress, which may increase susceptibility to anxiety, as suggested in previous studies (*Gunnar & Quevedo, 2007*).

In the current study, sleep quality was demonstrated to be significantly associated with anxiety, partially mediated by self-esteem and stress. Past research indicates that university students who have poor sleep quality are twice likely to be in anxiety as compared to those with better sleep quality (*Ramón-Arbués et al., 2020*), supported by several studies conducted in United States and Ethiopia (*Dagnew, Andualem & Dagne, 2020*; *Wilson et al., 2014*). Previous literature has predominantly focused on various aspects of mediating effects in psychological well-being and mental health disorders such as resilience and optimism (*Hesari & Hejazi, 2011*; *Kim & Jang, 2019*; *Lin, 2015*; *MengShi et al., 2015*; *Schwager et al., 2019*), however, there is lack of studies on the mediating roles of self-esteem or stress in the relationship between sleep quality and anxiety, even within the general population.

Low self-esteem has been linked to poor sleep quality, as evidenced by a 2013 study involving 1,800 adults, which revealed that individuals who slept six or fewer hours per night exhibited more pessimism and lower self-esteem compared to those who slept the recommended eight hours (*Lemola et al., 2013*). University students experiencing sleep

deprivation tend to become emotionally sensitive, prone to inattentiveness, and lack focus. Consequently, they may become less motivated to solve problems, even in situations that are typically manageable with sufficient sleep. As a result, they may experience a diminished sense of self-worth when confronted with minor criticism or errors due to their inability to effectively cope with such situations (*Stenson et al., 2021*). This can lead to feelings of apprehension, particularly concerning their ability to perform adequately in similar situations in the future.

Stress mediates the relationship between sleep quality and anxiety through the emotional dysregulation experienced by university students. Poor sleep quality has been associated with irritability, lethargy, and difficulty concentrating among stressed students (*Kalmbach et al., 2019*). In the present study, high mean sleep quality score (15.3 ± 2.5) may underscore the prevalence of sleep issues faced by undergraduates at UPM, particularly during examination periods. Emotional dysregulation due to sleep deprivation, often observed during late-night study sessions for exams, may contribute to increased anxiety levels (*Roth et al., 2006*). It has been suggested that sleep deprivation leads to elevated cortisol hormone levels during the day, potentially triggering anxiety (*Hirotsu, Tufik & Andersen, 2015*). Besides, sleep deprivation is linked to diminished function of brain pre-frontal cortex. Additionally, sleep deprivation is linked to reduced functioning of the brain's prefrontal cortex, which plays a crucial role in decision-making and coping with anxiety. Impaired function of this cortex may hinder effective decision-making regarding how undergraduates manage anxiety (*Goldstein et al., 2013*).

Previous literature has shown that university students with PIU are almost three times more likely to experience anxiety compared to those without PIU (*Ramón-Arbués et al., 2020*), supported by empirical evidence in a recent meta-analytic review (*Cai et al., 2023*). Past studies have investigated the role of self-esteem in mediating the relationship between PIU and various factors, such as subjective well-being and peer relationship (*Lin et al., 2023*; *Park, Kang & Kim, 2014*). However, there is a notable lack of empirical evidence regarding the mediating roles of self-esteem or stress in the relationship between PIU and anxiety. Similarly, the current literature on the role of stress in mediating PIU is also lacking. The present study reveals new findings indicating that self-esteem and stress played their roles in mediating the relationship between PIU and anxiety. The prevalence of PIU in the current study is high (85.3%). Further sub-analysis revealed that among all three domains studied, the mean score of control disorder was the highest compared to neglect and obsession. Internet use has been perceived as a mechanism for coping with deficiencies such as low self-esteem or high stress levels among students (*Widyanto & Griffiths, 2011*). The internet provides a virtual space for them to adopt different personalities and social identities, which can enhance their sense of well-being and satisfaction. Numerous studies have shown that young adults present themselves differently on social media platforms, dating apps, or online gaming platforms (*Poley & Luo, 2012*; *Prizant-Passal, Shechner & Aderka, 2016*). Additionally, online gaming, in particular, is among the most popular forms of internet use among young adults (*Widyanto & Griffiths, 2011*). It offers them a space to win, thereby increasing their sense of power and satisfaction. These feelings help improve their perceived deficiencies and positively evaluate themselves (*Pettorruso et al., 2020*). However, both of

these mechanisms tend to become repetitive and excessive as individuals perceive internet use as an escape from real-life problems such as university examinations and multiple academic tasks (*Lozano-Blasco, Robres & Sánchez, 2022*). Consequently, PIU may lead to procrastination, which in turn may cause feelings of guilt, shame, or inadequacy due to not meeting responsibilities on time. Moreover, persistent procrastination can instill a sense of losing control over academic duties, triggering anxiety about one's ability to manage tasks effectively. Ultimately, this cycle serves as a pathway to anxiety (*Anto et al., 2023*). Although individuals may acknowledge their PIU, they may struggle to control their internet use, as evidenced by the present study where control disorder scored the highest compared to neglect and obsession domains in PIU. This is supported by previous literature where addiction was positively linked with control disorder (*Bahrainian et al., 2014*). Therefore, this study highlights the importance of incorporating strategies to improve self-esteem and stress levels in line with interventional modules addressing PIU among undergraduates experiencing anxiety.

The majority of UPM undergraduates reported high social support (51.0%), with the association pathway between social support and anxiety in this study being fully mediated by self-esteem and stress. Previous research conducted in China found that university students with low perceived social support were almost six times more likely to experience anxiety compared to those with high perceived social support (*Ma et al., 2020*). However, the role of self-esteem or stress in mediating this pathway remains relatively understudied. Further analysis revealed a novel finding regarding mean score differences between support from significant others and from family or friends based on the level of social support perceived. Among undergraduates with high social support, the highest mean score for social support was from significant others compared to family or friends. Conversely, undergraduates with low social support scored the lowest for support from significant others, compared to family or friends. This finding illustrates that among students with similar levels of social support, the support received from significant others differs from that received from family and friends, highlighting the unique characteristics of perceived social support among UPM undergraduates. Additionally, a sub-analysis of the collected data revealed that the mean score of social support increased with the year of study. Further ANOVA analysis revealed that there is significant difference received within year of study; whereby the social support received by students from Year 1 and Year 4 differ significantly. Hence, junior students received significantly less social support compared to senior students.

This finding aligns with previous research, which found that high social support was among the determinants of high self-esteem (*Olga & Denrich, 2021*) or low level of stress (*Abdul Aziz, Baharudin & Alias, 2023*). As undergraduates enter university, they spend less time with family and friends and must adjust to a new, unfamiliar environment, which can be psychologically stressful. Social support from new friends at UPM may take time to develop, resulting in junior students perceiving lower social support compared to senior undergraduates. While family and friends primarily provide instrumental, financial, and informational support, emotional support from significant others may be a key difference related to anxiety in this study. This premise is supported by prior research emphasizing that emotional support is the type of social support most closely linked to mental health

conditions (*Yao, Zheng & Fan, 2015*). As adolescents transition to young adulthood, emotional support shifts from family and friends to significant others (*Zimmer-Gembeck, 2002*). Undergraduate students seek empathy, acceptance, intimacy, trust, and caring from significant others as their personal relationships evolve, which contributes to their sense of self-worth (*Kort-Butler, 2017*). Emotional support provides a safe space for them to express feelings of worry, loneliness, and apprehension when faced with stressful situations or demanding tasks at the university (*Yan, 2020*). Undergraduates with low self-esteem or in highly stressful situations may perceive low social support, especially in terms of emotional support, which can lead to anxiety, as demonstrated in the current study. Nonetheless, social support improves over the years, indicating the importance of strengthening social support among young UPM undergraduates.

The present study found that self-esteem fully mediates the relationship between MHL and anxiety among undergraduates, while stress does not act as a mediator in this pathway. There is a relative lack of research on the roles of self-esteem or stress in the relationship between MHL and anxiety, especially among undergraduates. Previous studies have only explored the role of different psychological mediators in MHL (*Qian et al., 2023*; *Xuemin et al., 2023*). Based on the results of the present study, UPM undergraduates exhibited the lowest mean scores in questions related to seeking help from psychologists or psychiatrists if they had mental illnesses (Item 4 and 16) (see Table S3). Poor help-seeking behaviour in mental health has been associated with low self-esteem in previous studies, through self-stigma. Undergraduates with low self-esteem may label themselves as socially unacceptable (*Corrigan, 2004*), which leads to negative self-perception and hinders them from seeking professional or non-professional help to protect their self-esteem when dealing with anxiety (*Nadler, 1986*). However, this avoidance coping mechanism is maladaptive and ineffective, contributing to the maintenance of anxiety (*Hofmann & Hay, 2018*). Therefore, incorporating self-esteem education for UPM undergraduates with anxiety is crucial in improving MHL.

Stress was hypothesized to mediate the relationship between MHL and anxiety based on the Biopsychosocial theory in the present study. However, stress did not play a role as a mediator as hypothesized. A sub-analysis of the current data showed that regardless of stress level, all students had good knowledge of stress, as evidenced by high scores in questions related to stress (Item 15) (see Table S3). According to general stress theory, problematic behaviour may arise due to stress (*Jun & Choi, 2015*). It has been suggested that those who receive less psychological training are more likely to have lower MHL, thus having a higher tendency for anxiety (*Wang & Lai, 2008*). Mental health services at UPM are primarily provided by the Counselling Division and Health Centre. These services encompass a range of offerings, including individual and group counselling, mental health assessments for students, medical treatment, and referrals to specialized care when necessary, particularly for anxiety-related issues. Additionally, the Counselling Division and Health Centre conduct numerous educational talks and mental health training programs aimed at enhancing undergraduate awareness and coping skills for mental health issues. In this study, we hypothesized that UPM undergraduates receive sufficient psychological training, particularly in stress management, enabling them to recognize and manage stress

effectively. As a result, when faced with challenges or stressful situations, they are adept at coping through various mechanisms, such as seeking social support, without experiencing excessive worry or fear.

## Recommendations

Based on the findings of the present study, it is recommended to develop an anxiety prevention policy that addresses factors such as sleep quality, problematic internet use, social support, and mental health literacy. This policy should prioritize self-esteem and stress as significant mediators influencing anxiety, particularly focusing on female undergraduates who were identified as a high-risk group. Improving sleep quality through practices like maintaining regular sleep schedules and reducing electronic device usage is crucial for reducing anxiety among students. Furthermore, interventions targeting undergraduates experiencing problematic internet use, such as cognitive behavioural therapy (CBT), educational programs, positive psychology, and multifamily group therapy, have been identified as effective (*Cañas & Estévez, 2021*). Notably, the present study found that undergraduates with problematic internet use reported the highest mean scores for control disorder symptoms, suggesting they could particularly benefit from CBT-based programs.

Prioritizing policies to enhance social support for undergraduates, especially emotional and financial support, is crucial. Strategies like peer support counselling and programs promoting social interactions can be integrated into the anxiety prevention policy to foster a supportive university environment where students feel connected and valued. In addition, incorporation of strategies 'on improving mental health literacy into the policy may also help in preventing anxiety among undergraduates. This includes incorporating mental health education into orientation programmes, integrating mental health topics into curriculum and establish peer education program, where trained undergraduate students deliver presentations, facilitate discussions, and promote mental health awareness campaigns across campus. Peers can often connect more effectively with fellow students and reduce stigma associated with seeking help.

The study underscores the pivotal roles of self-esteem and stress in mediating the biosocial factors contributing to anxiety. Implementing strategies to boost self-esteem, such as positive self-talk and mentorship initiatives, within the anxiety prevention policy can significantly benefit undergraduates. Positive self-talk involves developing affirmations and strategies to replace self-critical thoughts with constructive and supportive ones. Similarly, prioritizing existing stress reduction strategies at universities will further support efforts to mitigate anxiety levels among students.

## Strengths

This study is one of the pioneering efforts to explore how self-esteem and stress mediate the relationship between anxiety predictors among university undergraduates in Malaysia. Grounded in the Biopsychosocial model, this research expands upon previous studies by examining broader domains, enhancing our comprehension of this topic. The study achieved a high response rate and utilized a large sample size, employing probability

sampling techniques that allow for generalization to undergraduate populations across Malaysian universities.

Confidentiality and anonymity during questionnaire completion likely contributed to the high response rate and potentially more honest responses compared to face-to-face interviews. Furthermore, advanced statistical methods were employed to robustly analyse multiple factors simultaneously, revealing that certain predictors only attained significance when considering mediators. This underscores the critical importance of identifying and incorporating mediators in research endeavours.

### Limitations

This study, however, has several limitations that need to be addressed. It is cross-sectional, which means it cannot establish temporal relationships between factors, mediators, and anxiety. Several factors, such as prior mental health problems, academic performance, socioeconomic status, and diet quality, were not investigated due to time constraints. Additionally, the sample underrepresented male undergraduates, potentially limiting the generalizability of findings to this group compared to female undergraduates. Therefore, caution is advised when applying the results to male undergraduate populations.

The study is also subject to biases. The use of self-administered questionnaires introduces social desirability bias, particularly evident in responses related to sensitive topics such as alcohol consumption, illicit drug use, or self-esteem. Respondents may have underreported these behaviours or feelings due to concerns about stigma or judgment. To mitigate this bias, participants were assured of confidentiality and informed that their responses would not be disclosed to anyone, including parents, friends, or lecturers. Furthermore, questions about past activities, frequency, and intensity may lead to recall bias, as respondents may struggle to accurately remember details. These biases should be considered when interpreting the study's findings.

## CONCLUSION

The prevalence of anxiety among UPM undergraduates was relatively low compared to global and local estimates. This study demonstrated the significant role of self-esteem and stress in mediating biosocial determinants and anxiety. Interventional efforts targeting anxiety among undergraduates could significantly benefit from strategies aimed at enhancing self-esteem and mitigating stress. Future research should explore the levels of emotional social support and its association with anxiety among undergraduate students.

### Funding
The authors received no funding for this work.

### Competing Interests
The authors declare there are no competing interests.

## Author Contributions

- Muhammad Ikhwan Mud Shukri conceived and designed the experiments, performed the experiments, analyzed the data, prepared figures and/or tables, authored or reviewed drafts of the article, and approved the final draft.
- Anisah Baharom conceived and designed the experiments, performed the experiments, analyzed the data, authored or reviewed drafts of the article, and approved the final draft.
- Ahmad Iqmer Nashriq Mohd Nazan performed the experiments, analyzed the data, prepared figures and/or tables, and approved the final draft.

## Human Ethics

The following information was supplied relating to ethical approvals (i.e., approving body and any reference numbers):

Ethical approval was obtained from the Ethics Committee for Research Involving Human Subjects of Universiti Putra Malaysia (JKEUPM) (Reference number: JKEUPM-2023-164).

## Ethics

The following information was supplied relating to ethical approvals (i.e., approving body and any reference numbers):

Ethical approval was obtained from the Ethics Committee for Research Involving Human Subjects of Universiti Putra Malaysia (JKEUPM) (Reference number: JKEUPM-2023-164).

## Data Availability

The raw data is in SPSS format and can be found in the Supplementary Files.

## Supplemental Information

Supplemental information for this article can be found online at http://dx.doi.org/10.7717/peerj.19304#supplemental-information.

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
