# Peer review of "Self-esteem and stress: a structural equation modelling of biosocial determinants, psychological mediators and anxiety among Malaysian undergraduates"

_PeerJ, doi:10.7717/peerj.19304_

## Round 0.1 · original submission · Minor Revisions

You will see that both reviewers think highly of your work. Both have suggested several minor suggestions for improvement. In your resubmission, please include a rebuttal letter that goes through the reviewer feedback point-by-point and be very explicit in that document where edits have been made in the revised manuscript, or in cases where edits have not been made, what the rationale is for that.

From what I can see, the suggestions are minor and should be easy for you to address. I do however note one comment from R2, where they state "2. The authors did not describe all of the factors in the bio-social-psychological factors. There was not any explanation about it.". I don't think it is reasonable or necessary for you to talk about ALL factors in your introduction, as you did not examine all factors in your study. One way to address this comment could be to simply add a minor modification within introduction that there are many more factors that have been suggested to impact student anxiety (e.g., X, Y and Z), but for purposes of your research you have selected several of the well-established factors to examine.

·

Basic reporting

Overall, the paper is well-structured, self-contained, and clearly organized. The academic writing is polished and professional, presenting the information in a precise and effective manner. The figures and tables are well-designed and significantly contribute to conveying the results. An extensive body of literature and references is provided to introduce the background and support the findings.

Suggested Improvement:
In Table 4, it would enhance clarity to include brief explanations for Models 2-4 in the footnote to make the table more self-explanatory.

Experimental design

The overall study design is sound and clearly articulated. The use of Structural Equation Models (SEMs) and mediation analysis to explore relationships between observed items and latent factors is effective. I also commend the authors for utilizing multiple criteria to assess model performance, along with a sensitivity analysis. However, there are minor points that could be improved:

1. Lines 212-213: The statement "probability proportionate-to-size approach was utilized to select the required number of students from each faculty" is somewhat misleading. From my understanding, the probability proportionate-to-size (PPS) approach does not guarantee the selection of a required number of units from each group but rather defines the probability of each unit being selected based on its group size. I suggest revising this for more precise language.

2. Lines 395-399: The authors state, "Despite not meeting the typical AVE > r² criterion, the correlation coefficient (r) values for these five pairs of latent constructs were all below 0.9, ranging from 0.541 to 0.757. This aligns with criteria established by other researchers (Fornell & Larcker, 1981; Hair et al., 2010), thus confirming discriminant validity…". However, I was unable to locate the referenced criteria that indicate a correlation coefficient of less than 0.9 confirms discriminant validity. From my understanding of the literature, a correlation coefficient below 0.9 alone does not confirm discriminant validity. I recommend revising the explanation to reflect more accurate guidelines for confirming discriminant validity.

Validity of the findings

The data presented appears to be robust and of high quality. The conclusions drawn, particularly those related to association and mediation effects, are valid and well-supported by the data.

Additional comments

no comment

Reviewer 2 ·

Basic reporting

Dear editors and authors,
Thank you for the invitation to review the manuscript.
Thank you for the authors for writing the manuscript, reporting the study of anxiety in the context of mental health.
The following remarks are presented to make the manuscript better.
Thank you.

A. Introduction and Title

In general, the main issue of the study reported in the manuscript is anxiety, especially in the context of mental health.

Strengths:
1. It described the phenomenon of anxiety among university undergraduates, and its long-term consequences (37-49).
2. It presented several risk factors contribute to anxiety, namely biological factors, social factors, psychological factors, including the examples. They were followed by the description of some of the factors (50-58).
3. Jump to the recommendation: The authors wrote several issues to make an intervention (667-700), and the mental health service at UPM (653). It seemed that the study was conducted based on real problems.
4. (192-196) The authors wrote that the study would contribute to theoretical frameworks, namely the causal pathway or the mechanism of anxiety, by examining the mediating effects of self-esteem and stress.
5. It was written in the title that the study was carried out among Malaysian.
6. The authors described the meaning of several factors, such as sleep quality (74 - ), Problematic Internet Use (PIU) (87 -) etc., including the definition, the characteristics etc.
7. The conceptual model was presented on Figure 1.
8. The authors developed hypothesis (197) based on Biopsychosocial Theory

Weaknesses:
1. The terminology of anxiety has not yet been defined clearly in the manuscript, (for example DSM, ICD), the characteristics, etc.
2. The authors did not describe all of the factors in the bio-social-psychological factors. There was not any explanation about it.
3. The authors did not describe the real actual problems in the introduction.
4. Which one was the main focus of the study, the contribution to theoretical frameworks or the solutions of the real problems (see no 3).
5. The authors did not explain the reason why it should be among Malaysian.
6. The descriptions were not written consistently, for examples, it begins with the definitions, the characteristics, and followed by the issues related to the factors, etc.
7. The title was not as clear as the model on Figure 1. Revision of the title is needed.
8. The authors did not write the reference of Biopsychosocial Theory.

B. Methods

Study instrument
Strengths:
1. (226) it was written in the demographic data sheet, age, gender, ethnicity
2. It was described the instruments used in the study, such as SQS, PIUQ-6 etc.
3. There were descriptions of the scores and the meaning of the scores of the instruments.
4. (285- ) The questionnaire was provided in both English and Malay languages. It described the process of translation. There was pretest (292).

Weaknesses:
1. The authors did not write the source, on which this was based.
2. The authors did not write the authors of the instruments (who developed the instruments),
3. Not all of the scores were described. Not all of the psychometric properties of the instruments were presented, at least the Cronbach's alpha coefficient.
4. It was not very clear. The study used the English version of the Malay version of the questionnaires? Was the translation process carried out for all of the questionnaires?

Data collection and analysis
Strengths:
1. It was written “The questionnaires were distributed… (307)
2. (309) collected within a one-week period…
3. The analysis process was described

Weaknesses:
1. It was not written clearly, how the procedure of collecting data in detail.
2. How was it carried out in the period 19 June – 25 august 2023?

C. Results

Strengths:
1. (341- ) 1,193 undergraduates, 91.4%.
2. (345 - ) the mean age …
3. (367 - ) CFA stage …
4. (379 -) in the second stage…
5. The tables and figures were provided.

Weaknesses:
1. Was there any explanation why it was not 100%?
2. The source was not written. Was it Table 1? The content was a repetition of the table. There was any meaning of them? The data about ethnicity was not yet described.
3. The source was not yet written. Was it Table 2? What were the main findings of results presented on Table 2?
4. Where were the first stage, the third, the fourth?
5. The number of respondents were not yet written in every table and figure.

D. Discussion

Strengths:
1. (466 - 469) … significant difference in prevalence may be attributed to students' living arrangements
2. (494 - ) The study revealed that self-esteem and stress fully mediated the relationship between gender and anxiety among UPM undergraduates..
3. (503 - ) Our findings suggest that female undergraduates may be more sensitive to the effects of increasing cortisol levels…
4. Terminology: the authors sometimes used ‘relationship’ (e.g. 494,516, 530), sometimes ‘association’ (e.g. 507)
5. (543-544) Impaired function of this cortex may hinder effective decision-making regarding how undergraduates manage anxiety.
6. (653-665) It described about the facilities in UPM

Weaknesses:
1. Where was the supporting data for the explanation (students' living arrangements)? There was not any data in the demographic data sheet.
2. From which table the data was based on?
3. From which table the data was based on?
4. Which one was suitable for the study? If it described about prediction, it should be ‘association’.
5. What was the source?
6. There was not any explanation about the real problem (phenomenon) in UPM in the section Introduction.

E. Recommendations

Strengths:
1. The authors wrote about the possible interventions could be utilized based on the results.

Weaknesses:
1. It was not very clear, why it should be recommended. There was not any explanation in the section Introduction, that the study was aimed to give bases for developing intervention.

F. Abstract

Strengths:
1. The authors wrote about abstract.

Weaknesses:
1. Several issues should be first fixed in order to make revision in Abstract .

Experimental design

no comment

Validity of the findings

The comments are included in the no 1. Basic reporting

Annotated reviews are not available for download in order to protect the identity of reviewers who chose to remain anonymous.

---

## Round 0.2 · Minor Revisions

Before moving forward with the peer review process, we need to receive a proper response letter from the authors. In the response letter, for each response, please copy and paste text from the manuscript as quotes to make it clear what precise changes have been made and also include line numbers from the manuscript.

As one example, where the authors have stated a response as "The definition and characteristics of anxiety have been added". Alongside that response, please copy and paste the text from the manuscript that has been changed to effect the change that has been described, with line numbers from the manuscript where that text comes from. Do this for all responses in the response document.

---

## Round 0.3 · accepted · Accept

Thank you for making edits to the manuscript via taking into consideration the reviewer feedback.

Reviewer 2 ·

Basic reporting

The authors have revised the manuscript

Experimental design

Not applicable. The study did not use experimental design

Validity of the findings

The authors have revised the manuscript